# Influence of Tea Consumption on the Development of Second Esophageal Neoplasm in Patients with Head and Neck Cancer

**DOI:** 10.3390/cancers11030387

**Published:** 2019-03-19

**Authors:** Yao-Kuang Wang, Wei-Chung Chen, Ying-Ho Lai, Yi-Hsun Chen, Ming-Tsang Wu, Chie-Tong Kuo, Yen-Yun Wang, Shyng-Shiou F. Yuan, Yu-Peng Liu, I-Chen Wu

**Affiliations:** 1Division of Gastroenterology, Department of Internal Medicine, Kaohsiung Medical University Hospital, Kaohsiung 80708, Taiwan; fedwang@gmail.com (Y.-K.W.); jayshung1985@gmail.com (Y.-H.C.); 2Department of Medicine, Faculty of Medicine, College of Medicine, Kaohsiung Medical University, Kaohsiung 80708, Taiwan; 3Ph.D. Program in Environmental and Occupational Medicine, Kaohsiung Medical University, Kaohsiung 80708, Taiwan; tsubasawolfy@gmail.com; 4Research Center for Environmental Medicine, Kaohsiung Medical University, Kaohsiung 80708, Taiwan; hurley.lai@gmail.com (Y.-H.L.); mingtswu@gmail.com (M.-T.W.); 5Graduate Institute of Clinical Medicine, Kaohsiung Medical University, Kaohsiung 80708, Taiwan; 6Department of Physics, National Sun Yat-sen University, Kaohsiung 80424, Taiwan; ctkuo@mail.nsysu.edu.tw; 7School of Dentistry, College of Dental Medicine, Kaohsiung Medical University, Kaohsiung 80708, Taiwan; wyy@kmu.edu.tw; 8Department of Medical Research, Kaohsiung Medical University Hospital, Kaohsiung 80708, Taiwan; yuanssf@ms33.hinet.net; 9Center for Infectious Disease and Cancer Research, Kaohsiung Medical University, Kaohsiung 80708, Taiwan

**Keywords:** head and neck cancer, second esophageal neoplasm, tea, EGCG, arecoline

## Abstract

Alcohol is an important risk factor for the development of second esophageal squamous-cell carcinoma (ESCC) in head and neck squamous-cell carcinoma (HNSCC) patients. However, the influence of tea consumption is uncertain. We prospectively performed endoscopic screening in incident HNSCC patients to identify synchronous esophageal neoplasm. In total, 987 patients enrolled between October 2008 and December 2017 and were analyzed. In vitro studies were conducted to investigate the effect of epigallocatechin gallate (EGCG) on the betel alkaloid, arecoline-stimulated carcinogenesis in two ESCC cell lines. There were 151 patients (15.3%) diagnosed to have synchronous esophageal neoplasm, including 88 low-grade dysplasia, 30 high-grade dysplasia and 33 squamous-cell carcinoma (SCC). Tea consumption was associated with a significantly lower risk of having esophageal high-grade dysplasia or SCC in HNSCC patients, especially those who were betel nut chewers, alcohol drinkers or cigarette smokers (all adjusted odds ratio were 0.5; *p*-values: 0.045, 0.045 and 0.049 respectively). In vitro studies indicated that EGCG suppressed arecoline-induced ESCC cell proliferation and colony formation through the inhibition of the Akt and ERK1/2 pathway in a reactive oxygen species-independent manner. In conclusion, tea consumption may protect against the development of second esophageal neoplasms among HNSCC patients, especially those who regularly consume betel nuts, alcohol and cigarettes.

## 1. Introduction

Esophageal squamous-cell carcinoma (ESCC) is one of the most common second primary cancers occurring in patients with head and neck squamous-cell carcinoma (HNSCC) [1]. Development of second ESCC significantly compromises HNSCC patients’ survival [2,3]. Habitual alcohol, betel nut, and cigarette use are established risk factors for ESCC [4], but the influence of these carcinogens on the development of second ESCC in HNSCC patients is less clear. Studies using an endoscopic survey to identify second ESCC in HNSCC patients and to investigate possible risk factors, including alcohol, smoking, stage and location of index cancers showed some inconsistent results [3,5,6,7,8,9]. Ours was the largest study and identified alcohol drinkers, but not cigarette smokers or betel nut chewers, were at a higher risk to have second ESCC when first diagnosed as HNSCC [3]. In addition to these carcinogen exposures, tea consumption is another substance commonly discussed in the development of ESCC, but its anti-cancer effect remains inconclusive [10,11,12]. Our previous case-control study showed an inverse association between tea consumption and ESCC risk in a dose-response manner [13]. However, the influence of tea on the development of second ESCC in HNSCC patients has not been carefully investigated. 

Tea and its constituents have been studied in animals to prevent various cancers, including skin, lung, oral cavity, esophagus, stomach, liver, prostate, colon, bladder, and mammary gland [14]. Green tea, rich in polyphenols, has the potential to reduce cancer risks [15,16]. Epigallocatechin-3-gallate (EGCG), a major tea polyphenol, has been widely demonstrated in human cancer cell lines to inhibit carcinogenesis through its antioxidant activity, and its characteristics that suppress cell proliferation and angiogenesis, as well as increased cancer apoptosis [17]. However, whether polyphenols can counteract the effects of other carcinogens such as betel quid on developing ESCC is unknown. In this study, we investigated the influence of tea consumption on the development of esophageal squamous cell neoplasia among incident HNSCC patients. In vitro studies were also conducted to elucidate the effect of EGCG on arecoline-induced carcinogenesis in ESCC cells.

## 2. Results

### 2.1. Factors Associated with the Development of Synchronous Esophageal Neoplasm in HNSCC Patients

Initially, 990 HNSCC patients were enrolled in this study and only three (0.3%) were women; therefore, we excluded them in our analysis and focused on men. Most of the 987 HNSCC patients were habitual smokers (88.1%), alcohol drinkers (75.3%) or betel nut chewers (77.7%); 321 (32.5%) participants regularly drank tea. Through endoscopic screening, 151 patients (15.3%) were diagnosed to have synchronous esophageal squamous cell neoplasm, including 88 low-grade dysplasia, 30 high-grade dysplasia and 33 squamous-cell carcinoma (SCC) (Table 1). In total, 696 (70.5%) patients had stage III-IV HNSCC. Synchronous esophageal neoplasm was more commonly found in older patients (*p* = 0.019), smokers (*p* = 0.013), drinkers (*p* < 0.001), and those with advanced HNSCC (*p* = 0.029), as seen in Table 1.

In multivariate analysis, older age was significantly associated with the occurrence of esophageal low-grade dysplasia (adjust odds ratio (aOR) = 1.8; 95% confidence interval (CI) = 1.1–3.1), but not high-grade dysplasia/SCC (Table 2). For substance use, alcohol drinking was an independent risk factor for developing esophageal low-grade dysplasia (aOR = 3.4, 95% CI = 1.7–7.0) and high-grade dysplasia/SCC (aOR = 21.3, 95% CI = 2.9–156.6), while smoking and betel nut chewing were not. Tea consumers had a 50% lower risk of having esophageal high-grade dysplasia/SCC compared with non-consumers (aOR = 0.5, 95% CI = 0.3–0.9). Moreover, compared with stage 0–I HNSCC patients, those with stage IV diseases were 4.3-times more likely to have esophageal high-grade dysplasia/SCC (aOR = 4.3, 95% CI = 1.3–14.3) (Table 2).

We also examined the protective effect of tea against different substances on the risk of developing esophageal high-grade dysplasia/SCC, as seen in Table 3. Tea consumption reduced the risk of developing esophageal high-grade dysplasia/SCC by 50% among betel nut chewers (aOR = 0.5, 95% CI = 0.2–0.9), alcohol drinkers (aOR = 0.5, 95% CI = 0.26–0.99) and cigarette smokers (aOR = 0.5, 95% CI = 0.266–0.996). However, the protective effect from tea was not seen among non-users.

### 2.2. EGCG Suppressed Low-Concentration Arecoline-Induced Proliferation and Colony Formation of ESCC Cells

It has been shown that arecoline, the major alkaloid of the betel nut, promotes tumorigenesis of human oral squamous cell carcinoma (OSCC) cells [18,19]. A contradictory study showed that arecoline induced reactive oxygen species (ROS)-mediated apoptosis of OSCC cells [20]. To determine the effect of arecoline in ESCC cells, two ESCC cell lines, CE81T/VGH and OE21, which represented Asians (with betel nut chewing) and Caucasians (without betel nut chewing), were treated with a two-fold serial dilution of arecoline from 1000 μM to 3.9 μM for 72 h. High concentrations of arecoline elicited a cytotoxic effect on both cell lines and the half-maximal inhibitory concentration (IC_50_) of arecoline on CE81T/VGH and OE21 was 578.5 μM and 494.3 μM, respectively (Figure 1A). It is worth noting that low concentrations of arecoline, 15.6 μM and 31.2 μM, promoted the proliferation of CE81T/VGH and OE21 cell lines (Figure 1A,B). However, the treatment of 62.5 μM arecoline did not show a significant induction effect on cell proliferation of both cell lines and the cell viability decreased with the elevation in arecoline concentration.

To mimic the long-term arecoline exposure in areca nut-chewing ESCC patients, OE21 cells were maintained in the culture medium containing 15.6 μM arecoline for 14 days. Similar to the effect of the short-term treatment of arecoline, the prolonged exposure of low-dose arecoline significantly increased the proliferation of OE21 cells compared to the cells without arecoline treatment (Figure 1C). On the other hand, the treatment of EGCG reduced the arecoline-induced cell proliferation (Figure 1C). We next investigated the effect of arecoline on the anchorage and growth factor-independent growth, which represents the malignancy of tumor cells. The results from the soft-agar colony formation assay showed that the long-term treatment of arecoline significantly increased the colony number of OE21 cells, while EGCG reduced the arecoline-induced colony formation (Figure 1D, Appendix A).

### 2.3. EGCG Suppressed Arecoline-Induced Akt and ERK1/2 Phosphorylation Through An ROS-Independent Pathway

A number of studies have demonstrated that arecoline-induced cytotoxicity and changes of cellular physiology are mediated by ROS [21,22,23]. Most of these studies applied high concentrations (>300 μM) of arecoline in their experimental settings. However, the effect of low-dose arecoline on ROS production in ESCC cells is unknown. As shown in Figure 2A, the treatment of low concentration (15.6 μM) arecoline for 24 h was sufficient to induce ROS production in OE21 and CE81T/VGH cell lines compared to the vehicle control groups. Similar result was seen after treating OE21 cells with arecoline for 8 h (Appendix A). Consistent with the findings from others, EGCG or N-acetyl-L-cysteine (NAC) significantly reduced the arecoline-induced ROS production (Figure 2A).

Furthermore, to explore the molecular mechanism that may be involved in the arecoline-promoted cell proliferation, we focused on the activation of Akt and ERK1/2 signaling pathways, which are the oncogenic signaling pathways contributing to cell proliferation and tumorigenesis in different cancer types. As shown in Figure 2B, the treatment of low-dose arecoline (15.6 μM or 31.2 μM) transiently induced the phosphorylation of Akt and ERK1/2 at 60 min post-treatment in an OE21 cell line (Figure 2B, left panels). In a CE81T/VGH cell line, the arecoline-induced Akt and ERK1/2 phosphorylation occurred as early as 10 min post-treatment (Figure 2B, right panels). Thus, we hypothesized that the low-dose arecoline-induced Akt and ERK1/2 might be triggered by ROS. To our surprise, although EGCG inhibited the arecoline-induced Akt and ERK1/2 phosphorylation (Figure 2C), NAC failed to inhibit the arecoline-induced Akt and ERK1/2 phosphorylation (Figure 2D). These data suggested that arecoline induced Akt and ERK1/2 phosphorylation through a ROS-independent mechanism. In addition, the antioxidant activity of EGCG might not be required for its inhibitory effect on arecoline-induced Akt and ERK1/2 phosphorylation.

### 2.4. Inhibition of Akt Phosphorylation Suppressed Arecoline-Induced ESCC Cell Proliferation and Colony Formation

We next examined whether Akt or ERK1/2 phosphorylation was required for the arecoline-promoted proliferation of ESCC cells. Treatment of MK2206, a selective inhibitor of Akt1, Akt2 and Akt3, completely diminished arecoline-induced Akt phosphorylation (Figure 3A). The OE21 cells were pre-treated with a vehicle or low-dose arecoline for 14 days. Ten minutes before the proliferation assay, MK2206 was applied into the culture medium, and the cell proliferation was monitored by cell counting. Blockage of Akt phosphorylation by MK2206 reduced cell proliferation of the vehicle—and arecoline-treated OE21 cells (Figure 3B). On the other hand, the treatment of MK2206 also decreased arecoline-promoted colony formation (Figure 3C). We further examined the role of ERK1/2 phosphorylation on arecoline-promoted cell proliferation by using the selective ERK1/2 inhibitor, PD98059. Interestingly, although PD98059 decreased the arecoline-induced ERK1/2 phosphorylation (Figure 3D), it did not affect the cell proliferation and soft-agar colony formation of the vehicle and arecoline-pretreated OE21 cells (Figure 3E,F). Together, these data indicated that ROS-independent activation of the Akt signaling pathway, but not ERK1/2, is involved in low-dose arecoline-promoted cell proliferation of ESCC cells.

## 3. Discussion

Our clinical study showed that tea consumption might reduce the risk of developing second esophageal neoplasms in HNSCC patients, especially those who were substance users—suggesting tea constituents might attenuate esophageal carcinogenesis induced by betel nut, alcohol, and tobacco. Our in vitro studies further indicated that EGCG could attenuate the oncogenic property of ESCC cells after low-concentration, long-term arecoline treatment. Previous studies have reported a significantly inverse association between HNSCC risk and tea consumption, especially green tea [24,25]. However, the influence of tea consumption on the development of ESCC is uncertain and the findings are inconsistent, including the increased risk, no effect or inverse association, have been reported [13,26]. To the best of our knowledge, this is the first study to examine the effect of tea on the development of synchronous esophageal neoplasms in HNSCC patients. 

Tea is widely consumed in both Western and Asian countries and several mechanisms have been proposed for its anti-tumor effect, including its inhibition of c-Jun and ERK1/2 in lung cancer, phospho-Akt and nuclear β-catenin levels in colon cancer, and IGF/IGF-1R axis in colon and prostate cancer [27]. Some studies supported that tea polyphenol could counteract the effects of carcinogens from alcohol, tobacco, and betel nuts. For example, EGCG could reduce liver cell injury caused by oxidative stress from ethanol in vitro [28]. In cigarette-associated cell and animal studies, EGCG could suppress carcinogenesis of lung cancer and breast cancer by inhibiting nicotine-induced angiogenesis and epithelial-mesenchymal transition and α9-nicotinic acetylcholine receptors (nAChRs) signaling pathway, respectively [29,30]. For betel nut-associated carcinogenesis, arecoline-induced ROS in OSCC cells could trigger downstream effectors, including heat shock protein (HSP) 27, Snail, TGM-2, early growth response (Egr)-1, transforming growth factor (TGF)-β1 and Smad3 that further enhanced tumor progression [31,32,33,34,35]; some of which could be suppressed by EGCG [33,35]. Here, we demonstrated that arecoline-induced ROS could be significantly inhibited by NAC and EGCG. However, arecoline-induced Akt and ERK1/2 phosphorylation activation could only be inhibited by EGCG, not by NAC, suggesting a ROS-independent pathway that differs from previous OSCC studies [33,35].

Arecoline had been identified as a chemical ligand for M1, M2 and M3 muscarinic receptor (mAChR) and nAChRs in neuron, smooth muscle and coronary artery endothelial cells [36,37,38]. Binding of ligands with mAChR or nAChR leads to the activation of phosphoinositide-3-kinase (PI3K)/Akt and Ras/Raf/MEK/ERK1/2 signaling cascades. It has been shown that α9 and β2 subunits of nAChR were up-regulated in head and neck cancer, although the mAChR levels were not changed in normal and tumor tissues [39]. In addition, a previous study showed that betel quid extract activated a muscarinic M4 receptor-mediated ERK1/2 activation and promoted oral cancer cell migration [40]. Thus, we speculated that arecoline might function as a ligand for mAChR and nAChRs in the ESCC cell lines. This hypothesis might explain how arecoline activated Akt and ERK1/2 phosphorylation via an ROS-independent mechanism in our study. Further experiments are required to study whether the arecoline-induced Akt and ERK1/2 phosphorylation and cell proliferation are mediated by mAChR and nAChR.

In the present study, the time points of arecoline-induced Akt and ERK1/2 phosphorylation were different in two cell lines. For the CE81T/VGH cell line, arecoline exposure induced Akt and ERK1/2 phosphorylation in 10 min, and a second-phase Akt phosphorylation was observed after 60 min of arecoline treatment. On the other hand, Akt and ERK1/2 phosphorylation occurred after 60 min of arecoline incubation in an OE21 cell line. A previous study showed that the activation of mAChR led to PI3K/Akt and ERK1/2 phosphorylation in 5 to 10 min, whereas it took about 60 min to activate the nAChR-mediated signaling pathways in neuron and coronary artery endothelial cells [41,42]. Accordingly, we speculated that low-concentration arecoline may act as a ligand for mAChR in a CE81T/VGH cell line and for nAChRs in an OE21 cell line. 

In addition to the anti-oxidative effect, EGCG can bind to many intracellular proteins to regulate their functions [43,44,45,46]. For example, at physiological concentrations, EGCG is an ATP-competitive inhibitor of both PI3K and mammalian target of rapamycin (mTOR); thus, exerting its anti-cancer effect by inhibiting Akt phosphorylation and cell proliferation [45]. EGCG can also selectively inhibit multiple epidermal growth factor-dependent kinases to inhibit cell proliferation. It can inhibit ERK1/2 or AKT activity directly as well as through inhibiting the epidermal growth factor receptor (EGFR) activation [44]. Therefore, arecoline-induced Akt and ERK1/2 activation in our study may be suppressed by EGCG directly or through inhibition of other kinases, leading to decrease cell proliferation and colony formation.

There were several limitations in our clinical study. First, the information on lifestyle factors was collected by in-person interviews; thus, there could be recall bias and random misclassification, which might have decreased the significance of our findings. Second, the amount, subtypes (fermented or not) and temperature of the tea were not further analyzed because only 321 (32.5%) patients reported having a tea consumption habit. Tea is usually drunk at higher temperatures, but a protective effect was still observed in our participants. Third, induction chemotherapy or chemoradiotherapy for HNSCC may regress early esophageal cancer. However, we do not have complete data on the date of first chemotherapy although most of the participants received an endoscopy within two months of diagnosis. Finally, the status of human papilloma virus (HPV) infection, an important risk factor for HNSCC, was not examined in this study although the role of HPV infection for esophageal carcinogenesis was inconclusive.

## 4. Materials and Methods 

### 4.1. Study Population and Diagnosis of Esophageal Squamous Cell Neoplasm

Newly diagnosed HNSCC patients at Kaohsiung Medical University Hospital (KMUH) between October 2008 and December 2017 were prospectively enrolled for the endoscopic screening of esophageal neoplasm. The Institutional Review Board of KMUH approved this study (KMUH-IRB-980559) and all participants provided their written informed consent. The inclusion/exclusion criteria and endoscopic screening procedure have been described in our previous study [3]. Image-enhanced endoscopy, including narrow band image and Lugol chromoendoscopy, were performed on eligible patients within 6 months after diagnosis of HNSCC to detect synchronous esophageal neoplasm. This study excluded patients with other malignancies, those receiving prior esophageal surgery, those with total luminal obstruction caused by HNSCC, those needing emergent surgery for tumor bleeding or airway obstruction, those unsuitable or refused endoscopic survey and those with missing data of substance consumption, especially tea. We performed a biopsy on the suspicious neoplastic lesions and the definite diagnosis of esophageal neoplasms was confirmed by pathology. Esophageal squamous neoplasia was defined as low-grade dysplasia, high-grade dysplasia, and SCC in this study.

### 4.2. Substance Use and Demographic Data Collection

We collected participants’ demographic and lifestyle data by in-person interviews with questionnaires [3]. We defined alcohol drinkers as those drinking an alcoholic beverage at least once per week for a minimum of six months, cigarette smokers as those smoking ten cigarettes or more per week for at least six months, betel nut chewers as those chewing one betel nut, measured as quid, or more per day for more than six months, and tea consumers as those drinking tea at least once per week for a minimum of one year [13,47]. HNSCC stage was recorded according to the 7th edition of the American Joint Committee on Cancer (AJCC) tumor-node-metastasis system.

### 4.3. Esophageal Cancer Cell Lines and Chemicals

Two human ESCC cell lines, CE81T/VGH (Bioresource Collection and Research Center [BCRC] 60166) and OE21 (European Collection of Animal Cell Cultures [ECACC] 96062201), were used in the experiments. CE81T/VGH cells were cultured in Dulbecco’s Modified Eagle Medium (Cat: 11965-092, Thermo Fisher Scientific, Waltham, MA, USA), and OE21 cells were cultured in RPMI 1640 (Cat: 11875-093, Thermo Fisher Scientific) and then were kept at 37 °C in a humidified atmosphere with 5% CO2 incubator to maintain exponential growth. Both culture medium were supplemented with 10% fetal bovine serum (FBS; Cat: 1600044, Gibco, Waltham, MA, USA) and 1% antibiotic-antimycotic (Cat: 15240062, Gibco, Waltham, MA, USA), and additional 1% MEM non-essential amino acids solution (Cat: 11140035, Gibco) for CE81T/VGH. Arecoline was obtained from Thermo Fisher Scientific (Cat: AC250130250). EGCG, PD98059, and N-Acetyl-L-cysteine were obtained from Sigma Aldrich (Cat: 50299, P215, and A9165, St. Louis, MS, USA). MK2206 was obtained from Santa Cruz Biotechnology (Cat: sc-364537, St. Louis, MS, USA).

### 4.4. Cell Viability Assay and Cell Proliferation

The cytotoxic effect of arecoline was determined by cell viability assay. Briefly, the 2000 cells/well were seeded in 96-well plates and the cells were treated with arecoline at different concentrations for 3 days. After incubation, the culture medium in the 96-well plates was removed, and 100 μL of fresh culture medium and a pre-formulated 50 mL XTT mixed reagent (XTT reagent: electronically coupled reagent = 50:1) was added. The culture plate was incubated at 37 °C for 4 h. The absorbance of the samples was measured with a spectrophotometer at a wavelength of 475 nm. For counting cell proliferation, 2 × 10^4^ cells were seeded in six-well culture plates. The cells were harvested at different time points, and the cell numbers were counted using a hemocytometer.

#### DCFH-DA Cellular ROS Assay

For quantifying the level of ROS generating by arecoline, the OE21 cells were cultured in a 96-well plate with 5 × 10^4^ cell density per well that contains 50 μL culture medium. After 24 h incubation, we added 50 μL culture medium with 32 μM arecoline or 10 μM EGCG into each indicated well for 8 h. One mini moles H_2_O_2_ was used as ROS positive control. The warming 100 μL medium with 100 μM DCFH-DA (Cat: D6883, Sigma Aldrich) were added and incubated 30 min before harvesting cell. Cells were finally wash twice with warming serum-free RPMI 1640 and dissolved in 100 μL Dimethyl sulfoxide (Cat: D8418, Sigma Aldrich) that contained 1 mM NAC (Cat: A9165, Sigma Aldrich) for quenching reaction. After swirling for 10 s, 50 μL supernatant was transferred to black plate waiting for fluorescence evaluation. We read the fluorescence immediately with a fluorometric plate reader (BioTek FLx800 Microplate Fluorescence Readers, Winooski, VT, USA) at 480 nm/530 nm.

### 4.5. Soft Agar Colony Formation Assay

The soft agar assay was performed on six-well plates with the base layer of 0.5% agarose gel containing RPMI-1640 medium. To construct the cell layer, 5 × 10^3^ cells were suspended in 0.35% agarose gel with 10% FBS. The plates were incubated at 37 °C in 5% CO_2_ for 8–16 days to allow foci formation, and cell viability was determined by staining with 3-(4,5-dimethylthiazol-2-yl)-2,5-diphenyl tetrazolium bromide (MTT; 1 mg/mL). The colonies from each cell line were counted. All experiments were performed in triplicate, and the data are presented as the mean ± standard deviation (SD).

### 4.6. Western Blot

The OE21 and CE81T/VGH cells were treated with arecoline and/or EGCG and harvested and lysed in RIPA buffer (Cat: 9806, Cell Signaling, Danvers, MA, USA). The protein concentration was determined using a Pierce BCA kit (Cat: 23225, Thermo Fisher Scientific). Subsequently, 20 µg of total protein was loaded onto a 10% SDS-polyacrylamide gel for electrophoresis and transferred to a PVDF membrane (Cat: IPVH00010, Merck, Darmstadt, Germany). The protein was identified by incubating the membrane with primary antibodies for 18 h in 4 °C, followed by a horseradish peroxidase-conjugated secondary antibodies for 2 h in room temperature, and developed by West Pico PLUS Chemiluminescent (Cat: 34577, Thermo Fisher Scientific). The primary antibodies used in this study were listed as follows: anti-Akt (Cat: 9272), anti-phospho-AktT308 (Cat: 13038), anti-ERK1/2 (Cat: 9102) and anti-phospho-ERK1/2 (Cat: 9101) antibodies from Cell Signaling, and anti-actin (Cat: sc-47778) from Santa Cruz Biotechnology. The secondary antibodies were AffiniPure Goat Anti-Mouse IgG (H+L) (Cat: 115-035-003) and AffiniPure Goat Anti-Rabbit IgG (H+L) (Cat: 111-035-003) from Jackson ImmunoResearch laboratories (West Grove, PA, USA). The dilution ratios of all primary antibodies followed the manufacturer’s suggestions, and the dilution ratios of secondary antibodies were 1:10,000.

### 4.7. Statistical Analysis

The baseline characteristics of the study subjects were compared among three groups: no esophageal neoplasm, low-grade dysplasia, and high-grade dysplasia/SCC. The chi-square test was applied to analyze the categorized variables of clinical or demographic data and substance use, while the logistic regression model was used to evaluate the risks of substance use, including alcohol, betel nut, tobacco and tea, for the development of esophageal squamous cell neoplasm. The model was adjusted for age, substance use and cancer stage. The interaction between tea and other substances for the risk of high-grade dysplasia/SCC was assessed by logistic regression model after adjustment for age, cancer stage, betel nut, alcohol and tobacco. *p* value < 0.05 was considered statistically significant. All analyses were performed using the SAS 9.4 (SAS Institute Inc., Cary, NC, USA).

## 5. Conclusions

Our results suggested tea drinking could protect HNSCC patients from developing synchronous esophageal squamous neoplasm. Further cohort and in vivo studies are necessary to clarify the effects and mechanisms of different types and temperatures of the tea on the development of multiple upper aerodigestive tract cancers.

## Figures and Tables

**Figure 1 cancers-11-00387-f001:**
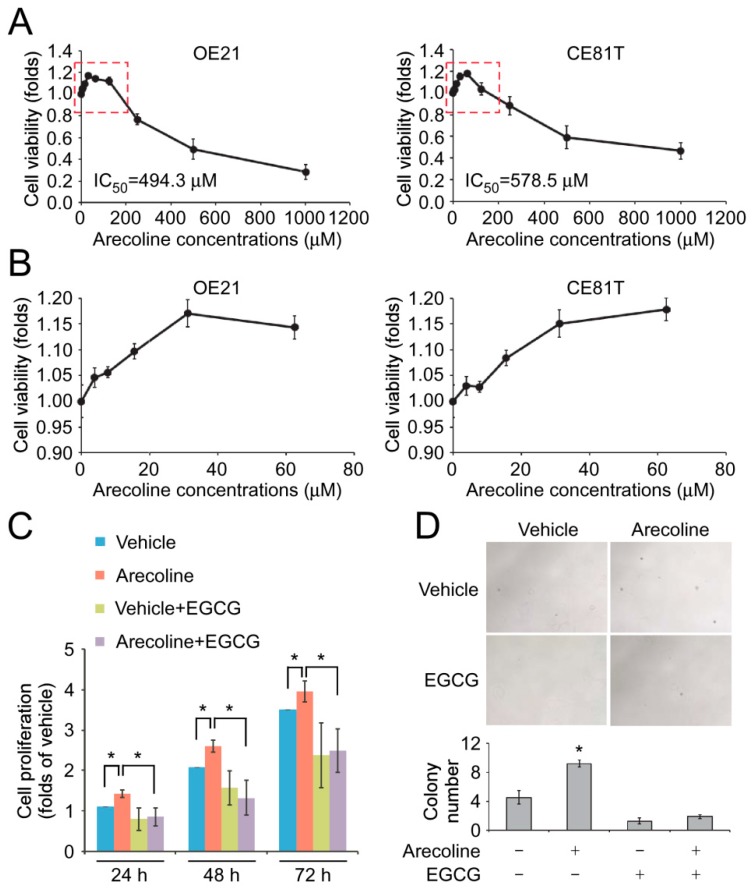
Epigallocatechin gallate (EGCG) reduced arecoline-promoted cell proliferation and soft-agar colony formation. (**A**) OE21 and CE81T/VGH cells were treated with different concentrations of arecoline for three days. The cell numbers were counted. (**B**) The data of low-dose arecoline (3.9, 7.8, 15.6, 31.2 and 62.5 μM) treatment acquired from the red-dot open box in (**A**). (**C**) OE21 cells were pretreated with 15.6 μM arecoline for two weeks, then the cells were treated with dimethyl sulfoxide (DMSO) (vehicle) or 15.6 μM arecoline with/without 5 μM EGCG for 24, 48 and 72 h. The cell growth was examined by cell proliferation. * *p* < 0.05. (**D**) OE21 cells were pretreated with 15.6 μM arecoline for two weeks, then the cells were treated with DMSO (vehicle) or 15.6 μM arecoline with/without 5 μM EGCG for 24, 48 and 72 h. The anchorage-independent growth of the cells was examined by soft-age colony formation assay. * *p* < 0.05 compared to the cells without arecoline and EGCG treatment.

**Figure 2 cancers-11-00387-f002:**
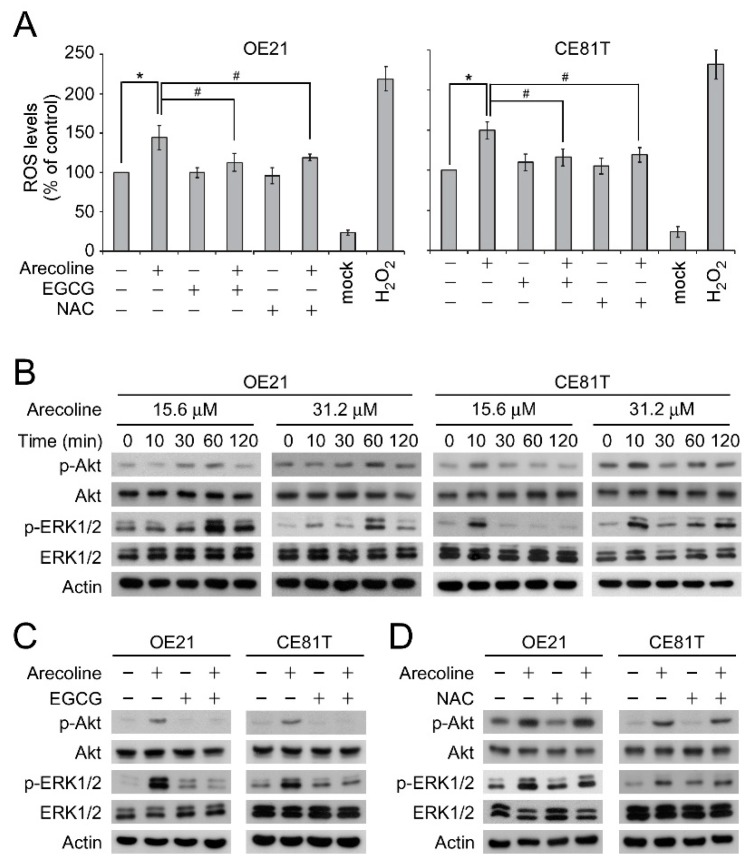
EGCG inhibited arecoline-induced Akt and ERK1/2 phosphorylation independent of its antioxidant activity. (**A**) ROS levels in the culture medium of OE21 cells treated with arecoline (15.6 μM), EGCG (5 μM) and/or NAC (10 μM) was examined. * *p* < 0.05, ^#^
*p* < 0.05. (**B**) OE21 and CE81T/VGH cells were treated with arecoline at indicated concentrations for different time points. The phosphorylation of Akt and ERK1/2 was analyzed by Western blot. (**C**,**D**) OE21 and CE81T/VGH cells were treated with arecoline and/or EGCG (5 μM) (**C**)/ NAC (10 μM) (**D**) for 60 min and 10 min for OE21 and CE81T/VGH respectively. The phosphorylation of Akt and ERK1/2 was analyzed by Western blot.

**Figure 3 cancers-11-00387-f003:**
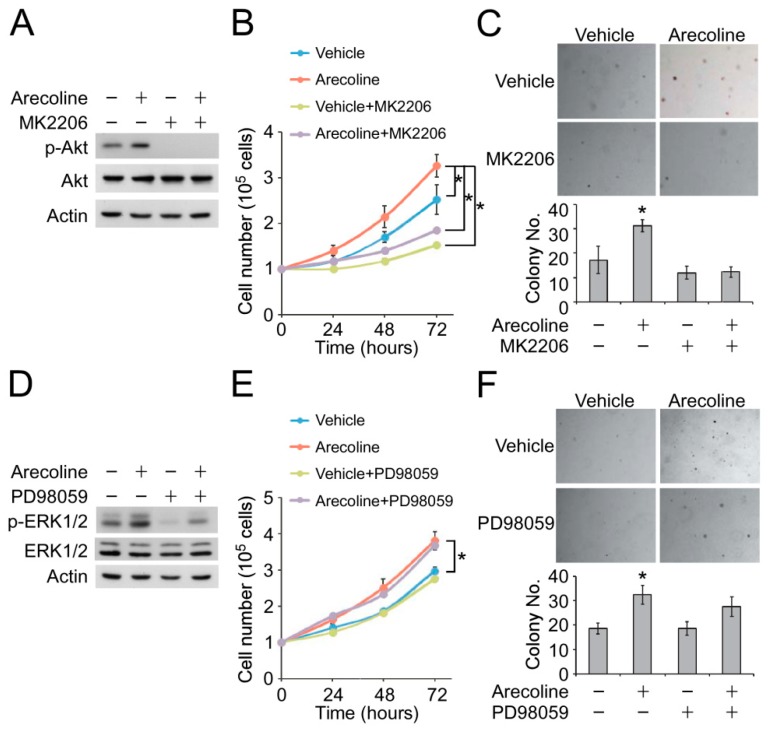
Arecoline-facilitated cell proliferation and colony formation were mediated by the Akt signaling pathway. OE21 cells were treated with arecoline and/or MK2206 (1 μM). (**A**) The Akt phosphorylation was examined by Western blot. The cell growth was analyzed by (**B**) proliferation assay and (**C**) soft-agar colony formation assay. OE21 cells were treated with arecoline and/or PD98059 (10 μM). (**D**) The Akt phosphorylation was examined by Western blot. The cell growth was analyzed by (**E**) proliferation assay and (**F**) soft-agar colony formation assay. * *p* < 0.05.

**Table 1 cancers-11-00387-t001:** Baseline characteristics of the 987 incident head and neck cancer patients.

Characteristics	No Esophageal Neoplasm (*n* = 836)	Esophageal Low Grade Dysplasia (*n* = 88)	Esophageal High Grade Dysplasia/SCC (*n* = 63)	*p* Value
	*n* (%)	*n* (%)	*n* (%)	
Age (years)				**0.019**
<50	279 (33.4)	20 (22.7)	28 (44.4)	
≥50	557 (66.6)	68 (77.3)	35 (55.6)	
Gender (Male)	836 (100)	88 (100)	63 (100)	
Smoking				**0.013**
Non-smoker	109 (13.0)	7 (8.0)	1 (1.6)	
Smoker	727 (87.0)	81 (92.0)	62 (98.4)	
Alcohol drinking				**<0.001**
Non-drinker	233 (27.9)	10 (11.6)	1 (1.6)	
Drinker	603 (72.1)	78 (88.6)	62 (98.4)	
Betel nut				0.953
Non-chewer	184 (22.0)	22 (25.0)	13 (20.6)	
Chewer	651 (77.9)	66 (75.0)	50 (79.1)	
Tea drinking				**0.025**
Non-consumer	562 (67.2)	53 (60.2)	51 (81.0)	
Consumer	274 (32.8)	35 (39.8)	12 (19.0)	
Cancer Stage				**0.029**
0–I	136 (16.3)	14 (15.9)	3 (4.8)	
II	117 (14.0)	15 (17.0)	6 (9.5)	
III	154 (18.4)	8 (9.1)	11 (17.5)	
IV	429 (51.3)	51 (58.0)	43 (68.3)	
Location of index cancer				**<0.001**
Oral cavity	623 (74.5)	55 (62.5)	17 (27.0)	
Oropharynx	117 (14.0)	19 (21.6)	15 (23.8)	
Hypopharynx	68 (8.1)	13 (14.8)	26 (41.3)	
Larynx	28 (3.3)	1 (1.1)	5 (7.9)	

Abbreviations: SCC, squamous cell carcinoma. *p* value was tested with Chi-square test.

**Table 2 cancers-11-00387-t002:** Risk factors for the development of esophageal squamous neoplasm in head and neck cancer patients.

Characteristics	Low Grade Dysplasia vs. No Neoplasm					High Grade Dysplasia/SCC vs. No Neoplasm				
	cOR	95% CI	aOR	95% CI	*p*	cOR	95% CI	aOR	95% CI	*p*
Age										
<50	1		1			1		1		
≥50	1.7	1.0–2.9	**1.8**	**1.1**–**3.1**	**0.03**	0.6	0.4–1.1	0.7	0.4–1.3	0.25
Smoking										
Non-smoker	1		1			1		1		
Smoker	1.7	0.8–3.9	1.7	0.7–4.0	0.22	**9.3**	**1.3**–**67.7**	5.6	0.8–42.2	0.09
Alcohol										
Non-drinker	1		1			1		1		
Drinker	3.0	1.5–5.9	**3.4**	**1.7**–**7.0**	**<0.01**	23.9	3.3–173.4	**21.3**	**2.9**–**156.6**	**<0.01**
Betel nut										
Non-chewer	1		1			1		1		
Chewer	0.8	0.5–1.4	0.6	0.4–1.1	0.09	1.1	0.6–2.0	0.6	0.3–1.1	0.08
Tea drinking										
Non-consumer	1		1			1		1		
Consumer	1.4	0.9–2.1	1.3	0.8–2.0	0.33	**0.5**	**0.3**–**0.9**	**0.5**	**0.3**–**0.9**	**0.03**
Stage										
0–I	1		1			1		1		
II	1.2	0.6–2.7	1.4	0.6–3.0	0.45	2.3	0.6–9.5	2.8	0.7–11.7	0.16
III	0.5	0.2–1.2	0.5	0.2–1.2	0.11	3.2	0.9–11.8	2.9	0.8–10.7	0.11
IV	1.2	0.6–2.2	1.1	0.6–2.1	0.69	**4.5**	**1.4**–**14.9**	**4.3**	**1.3**–**14.3**	**0.02**

Abbreviations: SCC, squamous cell carcinoma; cOR, crude odds ratio; aOR, adjusted odds ratio; CI, confidence interval. *p* value was tested with the logistic regression model for categorical variables for all the variables in the table.

**Table 3 cancers-11-00387-t003:** Influence of tea consumption on the development of severe esophageal neoplasm in head and neck cancer patients stratified by substance use.

Characteristics	No Neoplasm	High Grade Dysplasia/SCC	High Grade Dysplasia/SCC vs. No Neoplasm		
	*n*	*n*	aOR	95% CI	*p* value
Betel nut					
Betel nut non-chewer					
Tea non-consumer	123	10	1		
Tea consumer	61	3	0.8	0.2–3.3	0.705
Betel nut chewer					
Tea non-consumer	438	41	1		
Tea consumer	213	9	**0.5**	**0.2**–**0.9**	**0.045**
Alcohol					
Alcohol non-drinker					
Tea non-consumer	151	1	1		
Tea consumer	82	0	-	-	0.924
Alcohol drinker					
Tea non-consumer	411	50	1		
Tea consumer	192	12	**0.5**	**0.26**–**0.99**	**0.045**
Smoking					
Cigarette non-smoker					
Tea non-consumer	78	1	1		
Tea consumer	31	0	-	-	0.938
Cigarette smoker					
Tea non-consumer	484	50	1		
Tea consumer	243	12	**0.5**	**0.266**–**0.996**	**0.049**

Abbreviations: SCC, squamous cell carcinoma; aOR, adjusted odds ratio; CI, confidence interval. *p* value was tested with the logistic regression model for categorical variables for all the variables in the table.

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
