# Peer review of "Influence of Tea Consumption on the Development of Second Esophageal Neoplasm in Patients with Head and Neck Cancer"

_cancers, 2019, doi:10.3390/cancers11030387_

Round 1

Reviewer 1 Report

In this study, the authors analyzed the factors associated with the development of synchronous esophageal neoplasm in 987 HNSCC patients and 
investigated the influence of tea consumption on the development of esophageal squamous cell neoplasia among incident HNSCC patients. The results showed that tea consumption might reduce the risk of developing second esophageal neoplasms in HNSCC patients, especially those who were betel nut, alcohol, and tobacco users.

This study is interesting and the comments are as following:

1.    Did the authors check the HPV status of these HNSCC patients in this study since HPV status is a very important factor in HNSCC.

2.    It will be better if the authors could discuss more about why arecoline-induced Akt and ERK1/2 phosphorylation activation could only be inhibited by EGCG, not by NAC.

3.    The authors should confirm the in vitro results by in vivo results. Did the authors try in vivo experiments?

Author Response

1.     Did the authors check the HPV status of these HNSCC patients in this study since HPV status is a very important factor in HNSCC.

Response: Thank you for the comment. We did not check HPV status in our patients and have added this limitation in our manuscript (P9). Although HPV is an important risk factor for HNSCC, it might not be a confounder for the effect of tea on developing esophageal neoplasm in HNSCC patients. The role of HPV infection for esophageal carcinogenesis is still inconclusive.

2.     It will be better if the authors could discuss more about why arecoline-induced Akt and ERK1/2 phosphorylation activation could only be inhibited by EGCG, not by NAC.

Response: NAC is an anti-oxidant but EGCG may exert its anti-tumor effect by mechanisms in addition to the anti-oxidative effect. This was a phenomenon we observed in ESCC cell lines, and was different from the results in oral SCC as reported in other studies (References 33, 35). Therefore, we hypothesized that EGCG may counteract arecoline-induced AKt and ERK1/2 phosphorylation through a ROS-independent pathway. For example, EGCG may bind to intracellular proteins to regulate their functions (References 43-46). We discussed about how EGCG may inhibit Akt and ERK1/2 phosphorylation directly and indirectly in Discussion paragraph 5 (P8-9). EGCG may inhibit Akt and ERK1/2 indirectly by competitive binding on PI3K ATP pocket to stop downstream Akt activation [45], or through direct inhibition on Akt and ERK1/2 kinase function [44]. We have marked the relevant discussion in P8 in BLUE. Thank you.

3. The authors should confirm the in vitro results by in vivo results. Did the authors try in vivo experiments?

Response: Thank you for the suggestion. We don’t have the in-vivo data for green tea in esophageal cancer yet, but have established an arecoline-promoted esophageal tumor model in F344 rats (Oncotarget 2016 Dec 20;7(51):85244-85258). We can validate the effect of EGCG in this model. We have added this in the conclusion for future work. (p11)

Reviewer 2 Report

In this study, the authors investigated whether tea consumption reduce the incidence of second esophageal squamous cell carcinoma in HNSCC patients. Data from 987 patients were analyzed and the results showed that tea consumption was associated with lower risk of esophageal high-grade dysplasia or SCC in HNSCC patients. These results and the experiments conducted in this study are interesting to a wider audience. However, you need to address the following issues.

•Comments

1. You described upper endoscopic screening was performed within 6 months of HNSCC diagnosis. If the HNSCC patients underwent induction chemotherapy or chemoradiotherapy for HNSCC before screening endoscopy, their esophageal lesions were affected by the treatment. How many patients received upper endoscopic screening after any treatments for HNSCC? Why didn’t you define “within 6 months and before the start of HNSCC treatment”?

2. Although you defined tea consumers as “at least once per week”, the defined number seems to be a little small. How did you define that? What did you refer to?

3. I would like to see some results of the incidence of esophageal neoplasm according to the amount of tea consumption.

4. The software for statistical analysis need to be described in the text.

5. Generally, male patients predominate in head and neck cancer. However, it is a little strange that no female was included in 987 HNSCC patients. Your previous report included female patients (4.8%)(Reference 3). If you excluded female patients, you have to describe the reason.

I have found above mentioned issues that, once addressed, will improve the manuscript. In the present state, however, this manuscript is inappropriate for publication.

Author Response

1.     You described upper endoscopic screening was performed within 6 months of HNSCC diagnosis. If the HNSCC patients underwent induction chemotherapy or chemoradiotherapy for HNSCC before screening endoscopy, their esophageal lesions were affected by the treatment. How many patients received upper endoscopic screening after any treatments for HNSCC? Why didn’t you define “within 6 months and before the start of HNSCC treatment”?

Response: Thank you for your comment. We performed endoscopic screen within 6 months of HNSCC diagnosis because the definition of synchronous cancer was the diagnosis of second cancer within 6 months after primary cancer diagnosis. Actually, we performed endoscopic screen for most patients before cancer treatment. According to our previous study, 64% of the patients received endoscopic examinations within 4 weeks of HNSCC diagnosis and 90% of them within 68 days (Ref. 3). Those who received endoscopy later mainly received operation instead of chemotherapy or chemoradiotherapy. Therefore, HNSCC treatment may not substantially influence the screening result in this study.

2.     Although you defined tea consumers as “at least once per week”, the defined number seems to be a little small. How did you define that? What did you refer to?

Response: This questionnaire was modified from that used for our case-control study for esophageal cancer (Ref 13), we found tea consumption significantly reduced the risk of esophageal cancer by more than 50% in subjects who drank tea more than 1 time per week. The adjust ORs for esophageal cancer in subjects who drink tea 1-6 times per week and more than 7 times per week were 0.5 and 0.4, respectively compared with those less than once a week. One study on diet and upper-aerodigestive tract cancer in Europe (Ref 47: Int. J. Cancer 2009; 124, 2671–2676) used similar definition for tea consumption. We have added these two references to our definition of tea (Please see 4.2 in P9).

3.     I would like to see some results of the incidence of esophageal neoplasm according to the amount of tea consumption.

Response: We totally agree the dose-response data would add further information to our result. However, unlike alcoholic beverage which usually have the concentration of alcohol, the amount of tea polyphenol is difficult to measure even if we record the amount of tea consumption. The amount of tea leaf used in every preparation vary a lot and Taiwanese often drink iced tea. Moreover, only 321 (32.5%) patients reported having tea consumption habit and subgroup analysis was difficult here. We have added this to the limitation (Please see p9).

4. The software for statistical analysis need to be described in the text.

Response: We used SAS 9.4 and have revised the method in P11.

5. Generally, male patients predominate in head and neck cancer. However, it is a little strange that no female was included in 987 HNSCC patients. Your previous report included female patients (4.8%)(Reference 3). If you excluded female patients, you have to describe the reason.

Response: Yes, we exclude female patients when analysis because only 3 women (3/990, 0.3%) left after excluding participants by other criteria in this study. The percentage was kind of different from our previous work because more females did not have the information on tea consumption. We have revised this in our manuscript (Please see 2.1 in P2 and 4.1 in P9).

Reviewer 3 Report

Manuscript “Influence of tea consumption on the development of second esophageal neoplasm in patients with head and neck cancer” describes that tea consumption may protect against the development of second esophageal neoplasms among HNSCC patients. Paper is interesting. However, in my opinion, it fits better for more focused journals like MDPI Molecules.

The paper has a two parts, and both clinical and in vitro, could be presented as a separated manuscript.

In case of clinical part:

-          The inclusion/exclusion criteria should be presented in this study as well. Neither reader nor referee should not be forced to read another paper for that information.

-          I looked at the previous paper and the HNSCC patients are the one with oral cavity, oropharyngeal, hypopharyngeal and laryngeal cancer. In my opinion authors can’t put every tumour to the same group. Especially, when they using the cancer stage. Every single localization has its own TNM and staging. So analysis should be perform for all of them separately. The other thing is that authors did not exclude HPV-positive patients, or did not separate this group, especially in case of pharyngeal cancer. HPV is very important factor in pharyngeal cancer and such analysis will give new insight in the HPV-related cancerogenesis.

In case of in vitro part:

-          Fig 1D. The number of colonies is too small to make a conclusion.

-          I do not think that measuring ROS after 24h is reliable. In my opinion should be measured earlier.

Author Response

Manuscript “Influence of tea consumption on the development of second esophageal neoplasm in patients with head and neck cancer” describes that tea consumption may protect against the development of second esophageal neoplasms among HNSCC patients. Paper is interesting. However, in my opinion, it fits better for more focused journals like MDPI Molecules.

The paper has a two parts, and both clinical and in vitro, could be presented as a separated manuscript.

In case of clinical part:

-          The inclusion/exclusion criteria should be presented in this study as well. Neither reader nor referee should not be forced to read another paper for that information.

Response: Thank you for your suggestion. We added information on inclusion and exclusion criteria in section 4.1 in P9.

-          I looked at the previous paper and the HNSCC patients are the one with oral cavity, oropharyngeal, hypopharyngeal and laryngeal cancer. In my opinion authors can’t put every tumour to the same group. Especially, when they using the cancer stage. Every single localization has its own TNM and staging. So analysis should be perform for all of them separately. The other thing is that authors did not exclude HPV-positive patients, or did not separate this group, especially in case of pharyngeal cancer. HPV is very important factor in pharyngeal cancer and such analysis will give new insight in the HPV-related cancerogenesis.

Response: Thank you for the comment. We added the location of index cancer in Table 1 (P3). However, stratification by each primary site was difficult due to smaller number of patients for each cancer (also because of fewer tea consumers). We understand different index cancer has its own TNM system. Classification by stage was meant to examine whether esophageal neoplasm incidence would differ when HNSCC was detected in earlier or advanced/metastatic stages.

We did not check HPV status in every patient and have added this limitation in our manuscript (P9). Although HPV is an important risk factor for HNSCC, especially pharyngeal cancer, it might not be a confounder for the effect of tea on developing esophageal neoplasm in HNSCC patients. The role of HPV infection for esophageal carcinogenesis is still inconclusive.

In case of in vitro part:

-          Fig 1D. The number of colonies is too small to make a conclusion.

Response: Thank you for the comment. We performed the colony formation assay in triple replications, and each group contained twelve repeated measurements (Please refer to Supplementary Figure 1A). The trend of arecoline promoted colony formation was consistent in triple replications; same as the anti-proliferation ability of EGCG. In addition, the colony size with arecoline treatment was the biggest in all group, but shrank with additional EGCG treatment (Please refer to Supplementary Figure 1B). Therefore, from the trend of colony numbers and colony size, we concluded that arecoline promoted colony formation, which could be neutralized by EGCG.

-          I do not think that measuring ROS after 24h is reliable. In my opinion should be measured earlier.

Response: We used OE21 cells to establish DCFH-DA assay platform by treating OE21 with arecoline for eight hours (Please refer to Supplementary Figure 2). The results of ROS induction after eight-hour treatment was similar to that after 24-hour treatment in Figure 2. Thank you.

Round 2

Reviewer 2 Report

I would like to thank the authors for their appropriate answers. However, I have some considerations that should be taken in account.

 You answered “Those who received endoscopy later mainly received operation instead of chemotherapy or chemoradiotherapy. Therefore, HNSCC treatment may not substantially influence the screening result in this study.” However, it is very important issue whether patients included in this study received induction chemotherapy or chemoradiotherapy. I often experience that dysplasia and early esophageal cancer disappears after chemotherapy for head and neck cancer. The authors should show how many patients received chemotherapy before endoscopy. This should be added in the result and limitation.

Author Response

Thank you very much for the comment. We agree induction chemotherapy or chemoradiotherapy may regress early esophageal cancer. Since we don’t have complete data on the date of first chemotherapy and are not able to show how many of them received endoscopy before treatment, we listed this in the limitation. Please see the last paragraph of Discussion in P9.

Reviewer 3 Report

Authors improved manuscript according to my suggestions. Article can be accepted now for a publication in the journal.

Author Response

Thank you!